# The Ecological Separation of Deer and Domestic, Feral and Native Mammals in Tropical Northern Australia—A Review

**DOI:** 10.3390/ani14111576

**Published:** 2024-05-26

**Authors:** Peter J. Murray, Timothy D. Nevard

**Affiliations:** 1School of Agriculture and Environmental Science, University of Southern Queensland, Toowoomba, QLD 4350, Australia; 2The Cairns Institute, James Cook University, Cairns, QLD 4870, Australia; timothy.nevard@jcu.edu.au

**Keywords:** ecological separation, deer, introduced, feral, native mammals, herbivores, megafauna, tropical, northern Australian savanna, climate change

## Abstract

**Simple Summary:**

Simple Summary: In northern Australia, large native herbivorous mammals (weighing over 1000 kg) disappeared about 46 kya, and they have been replaced in the last 200 years by a range of introduced mammalian herbivores up to 1000 kg in bodyweight. Only one native herbivore has an adult bodyweight approaching 100 kg, and for the past 200 years, the total biomass of introduced domestic and wild vertebrate herbivores has massively exceeded that of native herbivorous species. Following a comprehensive review, we conclude that the current guild of native and introduced mammalian herbivores differentially utilises the landscape ecologically. However, climate- and anthropogenically related changes due to fire, drought, flooding, predation and introduced weeds are likely to have significant impacts on the trajectory of their relative ecological roles and populations. Given their differing ecological and dietary characteristics, against this backdrop, it is unclear what the potential impact of the dispersal of deer species could have in northern Australia. There is a dearth of supporting evidence to inform appropriate sustainable management should deer range expansion occur. We identify suitable research required to fill the identified knowledge gaps.

**Abstract:**

We explored the ecological and historical factors that led to formation of the unique guild of native and introduced mammalian herbivores between 5 and 1000 kg in northern Australia. Following the disappearance of large native herbivores about 46 kya, and until the arrival of Europeans and their livestock, the only herbivorous mammals were mid-sized endemic marsupial macropods, which continued to utilise the same vegetation as their much larger former neighbours. Only one species of contemporary native herbivore has an adult bodyweight approaching 100 kg, and for the past 150–200 years, the total biomass of introduced domestic and wild vertebrate herbivores has massively exceeded that of native herbivorous species. We conclude that the current guild of native and introduced mammalian herbivores differentially utilises the landscape ecologically. However, climate- and anthropogenically related changes due to fire, drought, flooding, predation and introduced weeds are likely to have significant impacts on the trajectory of their relative ecological roles and populations. Given their differing ecological and dietary characteristics, against this backdrop, it is unclear what the potential impact of the dispersal of deer species could have in northern Australia. We hence focus on whether sufficient knowledge exists against which the potential impacts of the range expansion of three deer species can be adequately assessed and have found a dearth of supporting evidence to inform appropriate sustainable management. We identify suitable research required to fill the identified knowledge gaps.

## 1. Introduction

Australia’s northern tropical savannas represent just under 25% of the continent (Figure 1). They consist of open forests, woodlands and grasslands with summer-dominant annual rainfall, where greater than 85% of rain falls in the wet season, followed by a dry season that is longer than 6 months. Their grassy understory comprises a wide diversity of native perennial C_4_ tussock grasses, and, in high rainfall areas (>1000 mm), annual native *Sorghum* species may dominate [1]. The woody stratum of Australian savannas is typically dominated by eucalypts (*Eucalyptus* and *Corymbia*), with extensive areas around the Gulf of Carpentaria dominated by the allied genus *Melaleuca*. In some systems, *Terminalia* are the dominant woody plants [1], with several million hectares dominated by the introduced prickly acacia *Vachellia nilotica* [2].

More arid systems are dominated by spinifex *Triodia* spp., which occur within a matrix of both chenopod and acacia shrublands. Sub-tropical systems comprise *Eucalypt*-dominated woodlands, with an understory of perennial C_4_ tussock grasses and increasing C_3_ grass components further southward. In the north-east, savannas fringe the fire-sensitive rainforests of the Wet Tropics, with a thin band of tall *Eucalypt* wet sclerophyll forest along its western boundary [1].

The abundance and species composition of woody vegetation in savannas is primarily determined by soils and climate [4,5,6]. Woody plant abundance, at local spatial scales, is regulated by fire and herbivory, and both alter vegetation structure and composition [5,7,8]. In this landscape, the role of herbivores and the complex set of factors that control herbivore abundance and plant defences is not well understood [9].

Australia’s savannas are characterised by an absence of large, mammalian herbivores [10]. There are only two widely distributed native herbivores greater than 5 kg in this landscape. These are the common wallaroo *Macropus robustus,* a mixed feeder, and the antilopine wallaroo, *M. antilopinus*, almost exclusively a grazer, both with mean densities less than 10 km^−2^ [11,12]. The common wallaroo only occurs at densities < 1 km^−2^ and is largely limited by dry season water availability, low forage quality, heat loads and predation [11]. Other macropods range from the agile wallaby *M. agilis* (<27 kg) up to the two largest extant species—the eastern grey *M. giganteus* (~65 kg) and the red kangaroo *M. rufus* (<90 kg).

Following the colonisation of Australia by Europeans, large ungulates were introduced from Asia, Europe and Africa. A total of eight species (and one size variant, the Timor Pony *Equus caballus*) have been introduced into various locations, and most are now widespread, so much so that the total biomass of native vertebrates is now far surpassed by that of non-native herbivores, especially cattle *Bos taurus* and *B. indicus*, as well as water buffalo *Bubalus bubalis*, camels *Camelus dromedarius*, pigs *Sus scrofa*, feral horses (brumbies) *Equus caballus* and donkeys *E. asinus* [13].

Some species of introduced herbivores, with both domestic and feral populations, are now widespread (e.g., cattle, pigs and horses); others have retained more localised distributions, such as the Bali banteng *Bos banteng*, and others, such as camels and donkeys, are still expanding their ranges. All of these are either entirely feral or primarily feral or maintain feral populations, together with minimally managed harvested livestock [14]. Other vertebrate species with adult liveweights of >5 kg include dingoes *Canis lupus dingo*, crocodiles *Crocodylus porosus* and *C. johnsoni* and a range of other introduced species, including relatively small numbers of deer: chital *Axis axis*, rusa *Rusa timorensis* and sambar *R. unicolor*.

Globally, tropical savannas typically support a range of megabrowsers [4], that is, species that predominantly browse, and mixed-feeding herbivores (species where 50% of their diet, on an annual basis, consists of woody plants and herbaceous dicots). In Africa, these include African Savanna elephants *Loxodonta africana* (2500–6000 kg), the white rhinoceros *Ceratotherium simum* (2000–3600 kg), the black rhinoceros *Diceros bicornis* (700–1400 kg) and giraffes *Giraffa camelopardis* (800–1200 kg), plus other smaller mixed-feeding and browsing mesobrowsers that vary in body size, diversity, abundance and feeding niches. These preferentially feed on woody plants and are mostly <500 kg but range from about 5 kg (e.g., dik-dik, *Madoqua* spp.) to 1000 kg (e.g., eland, *Taurotragus* spp.). Until their extinction prior to 46 kya, Australia had a diverse megafauna, with 20 or more genera of giant marsupials, including marsupial browsers and mixed feeders, including the 3.7 m/2700 kg generalist *Diprotodon optatum* [15], <230 kg Sthenurine kangaroos and the ~500 kg browsing “marsupial tapir” *Palorchestes azeal*, as well as large flightless birds and reptiles [16]. As this range of species became extinct, there have been no native megabrowsers in Australia for around forty-six thousand years [17].

Humans were settled in Australia by 65 kya and were widespread across the continent by about 46 kya [18,19]. Today, Australia has no native animal heavier than 100 kg (i.e., all native grazing, browsing or intermediate herbivores are mesobrowsers or smaller), but as stated above, for most of the Pleistocene, Australia supported a rich assemblage of much larger vertebrates. Climate change and human-induced vegetation change caused by anthropogenic burning have been considered possible causes of the extinction of the Australian Pleistocene megafauna [1]. However, multiple independent lines of evidence point to direct human impact as the most likely cause of the extinction of the large vertebrates by about 46 kya [19,20,21].

A number of theories suggest the paucity of native Australian vertebrate megabrowsers had a negligible functional role in relation to vegetation [22], but this has been disputed by others, for example, Rule [16] and Cook [1] and their colleagues. At least three large, introduced herbivores, water buffalo, banteng and cattle, may occupy the feeding niches left vacant by the Pleistocene extinctions of the marsupial megafauna [10,23], although there is some evidence of competition between introduced bovines and native herbivores [23], but this does not mean contemporary habitats are necessarily unsuitable for large grazing or browsing mammals. For example, “over two-thirds of the vascular plant genera present in the Northern Territory between 11 and 16° S have global distributions” [14] (p. 445). Most, if not all, contemporary species of native plants would have been present at the time of the megafauna and hence subject to their grazing or browsing. This caused Cook and colleagues [1] (p. 55) to ask, “Are Australian savannas the ghost of a system that lost its browsers 46,000 years ago?”.

In locations where their populations have been established in Australia for >50 years, the densities of feral species can be comparable to those in their native habitats. With a mean biomass of 2225 kg km^−2^, introduced and feral ungulates in parts of northern Australia have a similar biomass to that of ungulates in some savanna regions globally [14,24]. Some research has indicated that the removal of grazing pressure by introduced herbivores may help restore populations of native mammals [25], but this might not occur where local vegetation has been permanently impacted, as it has over much of northern Australia, or where small mammal populations have been severely reduced [26] due to predation by feral cats *Felis catus* [27,28].

Habitat change associated with domestic or feral animals is often assumed to be unusual or undesirable. Such undesirable effects may include loss of habitat for native species and soil erosion [29,30]. However, it is open to question whether the impact of domestic and feral herds is in any way greater than or in some way different from that which occurs in the savannas of Asia or Africa [14,31] or which occurred in Australia prior to the extinction of the megafauna [31]. If the impact of domestic and feral herds on savanna vegetation in contemporary northern Australia is greater than occurs with the much lower post-megafaunal native herbivore biomass, one possible explanation is that feral herds might exist at densities greater than those achieved in their native habitats. Freeland [14] indicated that all 10 introduced herbivorous species for which there are data had ecological densities greater than those predicted by the Damuth relationship (i.e., an inverse relationship between the size of an animal species [herbivores] and its local abundance). Possible explanations for the unusually high population densities of introduced herbivores in the Australasian environment include (a) the absence of competition from species-rich herbivore communities; (b) a paucity of potential predators; (c) a paucity of parasites and diseases and (d) an absence of allelochemical/physical defences capable of protecting Australasian plants from introduced herbivores. Moreover, there is some evidence that the introduced herbivores in northern Australia lack significant impacts from predators and pathogens [14].

The purpose of this review is to identify the extent of research into the ecological context in which deer and domestic, feral and native mammals coexist and to identify gaps in the research that could facilitate appropriate management of expanding deer populations in northern Australia. We therefore adopted the following methods.

## 2. Materials and Methods

### 2.1. Review Process

A review of the published literature assessed studies of the diet, digestive physiology, habitats and ecological separation of ungulates, with a focus on three species of deer (chital, rusa and sambar), as well as camels, horses, donkeys, cattle, water buffalo, macropods and pigs, in tropical northern Australian landscapes and other parts of the world. A pivotal area for the review was what changes to this guild of herbivores, including their current environmental impacts, could follow the dispersal of populations of chital, rusa and sambar across northern Australia. This required the extraction of information on the ecological roles of all contemporary northern Australian herbivores and their environmental impacts. When available, limitations and recommendations for future research were also recorded from the literature. The recommended additional research is therefore based on suggestions in the literature and the ideas of the authorship team.

### 2.2. Selection Criteria and Search Databases

Searches of peer-reviewed articles were made using the Scopus, Google Scholar and Web of Science academic databases and the search function in the journals *Small Ruminant Research* and *Wildlife Research* from May to July 2023. A text string search of (“ecological separation” OR “habitat separation” or “digestive physiology” or “dietary overlap”) AND (“ruminant” or “cattle” or “pig” or “camel”) AND (“deer” or “chital” or “rusa” or “sambar”) was made for a combination of article titles, abstracts and keywords. Full texts were sourced via the University of Southern Queensland and the University of Queensland and supplemented with Scopus, Google Scholar and other web-based searches.

### 2.3. Exclusion Criteria

Selection criteria were developed prior to the search, with exclusion criteria for studies including: (i) ex situ diets fed to focal animals (e.g., zoo observations); (ii) papers not written in English; (iii) the re-use of data from other published studies; (iv) did not specify how animals’ diet or behaviour were observed (insufficient methodology) and (v) did not differentiate animal to species level. No studies were excluded from the literature review synthesis due to publication date. The search returned a large quantity of relevant articles (n = 800).

## 3. Results

In tropical savannas, mesobrowser abundance (i.e., browsers, grazers and intermediate feeders) varies markedly between continents, with native mesobrowsers rare in Australasian savannas [4]. Compared to tropical savannas in Asia and Africa, northern Australian savannas are depauperate in large native herbivorous mammal species. For example, in the Northern Territory, there are only six macropod species that could qualify as mesobrowsers. Asian and African savannas have at least six to eight species of native mesobrowser [14], but in northern Australian savannas, it is unusual to find more than four living in the same geographical area (Table 1 and Figure 2), and even then, they appear to exhibit different patterns of habitat choice [14].

From Table 1, the diets and habitat requirements of native macropods appear to indicate potential competition as follows:i.Between agile wallabies, antilopine wallaroos and common wallaroos, and where black wallaroos occur in the Northern Territory. These species have some level of dietary overlap (they are all grazers but may include other foods, e.g., fruits) but display habitat separation: black and common wallaroos are found in “rocky hills”, whereas the agile wallaby prefers open forest, and the antilopine wallaroo prefers tropical woodlands.ii.Agile and northern nail-tail wallabies, antilopine wallaroos, common and black wallaroos, black and whiptail wallabies and red and eastern grey kangaroos have similar diets. They are all grazers, except the black wallaby, which is primarily a browser, and the northern nail-tail wallaby, which mostly eats herbs but may also include other foods, e.g., succulents or fruit. Black wallabies will eat some exotic and poisonous plants.iii.Red and eastern grey kangaroos and the common wallaroo are predominantly grazers, although they utilise different habitats: respectively, open plains, open forest and rocky hills.

The distributions of introduced mammals > 5 kg in northern Australia (Figure 3) indicate several overlaps in their distributions with those of native mammal species, shown in Figure 2. Figure 3 does not show the distribution of feral horses or cattle (both grazers), as they are distributed across much of northern Australia and as such overlap with most other introduced and native grazing species. The distributions of the introduced buffalo, donkeys and feral pigs, with different diets (Table 2), but some overlap during drier periods, and habitat requirements, overlap with those of agile wallabies and antilopine wallaroos but not common and black wallaroos *M. bernardus,* where they are found in the Northern Territory (Figure 2).

There is also potential dietary competition for grasses between donkeys and feral horses, cattle, banteng, feral pigs, agile and northern nail-tail wallabies *Onychogalea unguifera*, antilopine wallaroos, common and black wallaroos and red kangaroos, plus black wallabies *Wallabia bicolor*, whiptail wallabies *M. parryi* and eastern grey kangaroos. In more arid regions, there is potential competition between camels (although they are preferential browsers; in drought, they eat grasses) and grazers and the northern nail-tail wallaby (although they focus on herbs) and the common wallaroo. In most of the south-western parts of the Northern Territory and central WA, camels are the only introduced species that could potentially compete with red kangaroos and common wallaroos.

Compared to other tropical savannas, there is an absence of native megaherbivores in Australia, where the largest native macropod species has a mature bodyweight < 100 kg (Figure 4), with all 10 introduced species having higher bodyweights, 7 of which range between 2.5 and 8 times that of the largest native species. Of the introduced herbivores, the camel is a pseudo-ruminant and a desert-living preferential browser; banteng and water buffalo are both grazers and have a strong dietary preference for sedges; donkeys and feral horses utilise hindgut fermentation and, along with ruminant domestic and feral cattle, are also grazers. The mature bodyweights of mammals > 5 kg in northern Australia were graphed in Figure 4 below, emphasizing the massive size difference between the remaining native and introduced herbivores in tropical Australia.

Considering the large and diverse range of herbivores much larger than 100 kg in Africa, such as elephants, rhinos, giraffes, hippopotamuses *Hippopotamus amphibius* and buffalo *Syncerus caffer,* and South Asia, such as elephants *Elaphas maximus*, rhinos *Rhinoceros unicornis*, gaur *Bos gaurus* and water buffalo, the post-megafaunal absence of equivalent large-sized mammals in tropical Australia is striking. There has been contemporary discussion that the presence of recent herbivore arrivals in Australia shown in Figure 4 is potentially returning and/or replacing the ecological roles formerly occupied by the extinct megafauna. Although there is limited information about the mechanisms according to which these species co-existed, it is likely to involve a combination of ecological separation and at least some differences in their diets, digestive physiology and feeding behaviour [17,31]. 

At present, there is relatively poor representation across large regions of northern Australia of its six introduced cervid species. However, as three of these species—chital, rusa and sambar—are endemic to climatic regions not dissimilar to much of tropical northern Australia, it is not unreasonable to expect that the small populations that exist in Queensland and the Northern Territory [33,34] (Figure 3) could disperse, given that other introduced species, e.g., bovids, equids, camelids and pigs, have successfully dispersed across these landscapes.

Although they are potentially well adapted and their range potential could be significant, the current and potential ecological roles of deer in northern Australia have yet to be comprehensively investigated. Climatch analysis ([35] using an algorithm that predicts the likely range of an exotic species by comparing climates in occupied and potential locations) (Figure 5) indicates very wide potential distributions for chital, sambar and rusa [35]. However, some populations have existed close to their original points of release for over 150 years. Given their differing external and gastroenteric morphology, dietary and habitat preferences and ecological roles, their dispersal is likely to be very different from the crude predictions of Climatch and their impacts on introduced and native biota different from those of other introduced herbivores. The relationships between introduced deer and the rapidly changing ecological situation in tropical savanna ecosystems (due to climate change, floods and drought, shifting botanical composition, increasing grazing pressure and altered fire regimes) are likely to be complex.

As a cautionary note, the bioclimatic maps of potential deer distribution in Moriarty [34], and more recently from Davis and colleagues [35], indicate that chital, rusa and sambar are capable of establishing populations across northern Australia, including in regions where currently droughts and fire are drastically reducing both native and introduced species. However, given the increasing frequency and intensity of fires, droughts and floods, it is difficult to see how deer could disperse this widely, let alone reach and maintain the predicted high populations.

In South Asian savannas, from which the deer in Australia’s tropics mainly originate, the six most common mesobrowsers are chital, sambar, nilgai (*Boselaphus tragocamelus*), chinkara (*Gazella bennetti*), chousingha (*Tetracerus quadricornis*) and muntjac (*Muntiacus vaginalis*). Of those that have been well studied, chital may consume > 90% grass during wet seasons [36,37] and could be affected by competition with cattle for grass forage [38,39]. This indicates chital depend on grass as an important component of their diet [4]. In contrast, other studies report less consumption of grass (30–70%) by chital during the wet season and substantial increases in browse in their diet during the transition from the wet to dry season and in the dry season [40,41]. Furthermore, Nepalese savanna chital are highly selective browsers, and they consume 31 woody species, 15 of which are consumed more than would be expected based on availability [37].

Based on their digestive morphology, Hofmann [42] placed chital as intermediate/mixed feeders rather than grazers, although empirical evidence indicates that they may eat more grass than mixed feeders, such as the African impala *Aepyceros melampus*. The relatively high densities of chital in central–east Queensland [43,44,45,46], combined with their preferential consumption of woody plants during wet–dry transitions and the dry season, could therefore imply competition with Australian native and domestic grazers, unless their foraging is temporally, spatially or behaviourally separated. Sambar, also mixed feeders, consume more browse (50–90%) compared with chital [40,47,48]. 

Grass and browse represent very different food resources, and their consumption poses different constraints for herbivores [49]. Browse species, when compared to grasses, have higher levels of soluble cell content and nitrogen, which are beneficial for large herbivores but at the same time have higher levels of lignin and secondary metabolites, which can be detrimental [49,50,51,52]. Macropods have a distinctive feeding characteristic, different from all antelope, deer, goats, camelids and giraffes, in being able to use their forelimbs to grab and manipulate branches, a factor in habitat suitability that needs to be better understood in relation to competitive niche occupancy [1].

These fundamental differences between grasses and browse have led to different adaptations amongst species specialising in one plant type or the other, with implications for all aspects of their ecology [49,51,53,54,55,56]. Based on the differences found in the relationships between the stomach structures and feeding habits of East African ruminants, Hofmann and others proposed that species’ digestive systems were the primary factors deciding forage selection and feeding habits [53,54,56]. Subsequent analyses that statistically accounted for differences in body mass found limited evidence for morphological and anatomical differences between large herbivores belonging to different feeding categories [51,52]. However, further studies have found evidence to support Hofmann’s basic proposition [57,58,59], although body mass clearly plays a role in the resource ecology of large herbivores [60]. Hofmann and colleagues [56] originally based their foraging categories on differences between ruminant species, foregut fermenters, in the order Artiodactyla, as 92% of the 260 large herbivore species worldwide are ruminants. The remaining 8% of large herbivores are hindgut fermenters, in the orders Perissodactyla and Proboscidea.

An interesting perspective on the loss of large native herbivores, and their subsequent “replacement”, is given by Lundgren [17] and others, who postulated that introduced herbivores “may, in part, restore ecological functions reflective of the past several million years before widespread human-driven extinctions” [17] (p. 7871). In Australia, the timescale of these human-driven extinctions may have occurred much more recently (i.e., within the last c. 46,000 years) as two waves, the first following the arrival of indigenous people and the second following the arrival of Europeans. Introduced species (both domestic and wild), given their ability to consume large quantities of vegetation, have the capacity to influence ecosystem processes such as wildfire and shrub expansion. Lundgren and his colleagues also make the point that “most extant plant and animal species evolved in the context of diverse large-bodied herbivore assemblages from the early Cenozoic (30–40 million ybp) until the Late Pleistocene extinctions” [17] (p. 7871) and that introduced herbivores in Australia have numerically replaced lost species richness by about 50% [17] (p. 7872).

## 4. Discussion

Competition and displacement theory suggests that unless an introduced species has some form of competitive advantage (e.g., access to and the ability to use an abundant resource not available to other species), it is unlikely to establish within a natural ecosystem. The presence of a significant number of introduced herbivores across northern Australia and native herbivorous species, plus the growing frequency of environmental and climatic disturbances (https://www.ipcc.ch/report/ar6/wg2/ (accessed on 23 April 2024)), suggests that deer may be potentially most able to establish populations in locations with minimal environmental/climatic disturbance and where there are effectively empty niches for them. From studies of their native ranges and physiology, we know that chital, rusa and sambar are differentiated in their dietary and ecological requirements [42], and there are populations of chital near Charters Towers in Queensland that appear to have been limited in their dispersal [46].

Although we do not know whether this is true for deer, we do know that in large regions of northern Australia, phosphorus supplements are required for successful commercial cattle production [46]. Following on from this are the unknown impacts of ruminant pathogens (i.e., diseases and parasites) and the pathogens of other mammals, plus the potential impacts of crocodile and dingo/wild dog predation. In parts of Australia where dingoes are allowed to continue to fulfill their role as apex predators, they are thought to predate heavily on deer, particular fawns (e.g., red deer *Cervus elaphus* in south-east Queensland and chital near Charters Towers), as well as goats (particularly kids). Thompson and colleagues [61], working in High Country Victoria, found that dingo diets typically comprised up to 44% sambar.

We know that sambar, of tropical (Sri Lankan) provenance, have been exceptionally adaptable in Australia, living in snow in Victoria and dispersing through the alpine environment in NSW, thence northwards into milder regions. On the other hand, in northern Australia, sambar have hitherto been confined to an isolated population in the Cobourg/Garig Gunak Barlu National Park. Given their native range, they would appear to have the potential to thrive in northern Australia, so it is mystifying why, similarly to banteng, sambar have not appeared to have spread historically from the Cobourg peninsula. Demystifying this through targeted research would therefore be instructive in predicting the constraints on the dispersal of sambar and other deer species in tropical Australia.

The effects of increasingly frequent extreme floods, fires and droughts; predation from dingoes/wild dogs and crocodiles and management by humans (e.g., hunting, shooting and baiting) all have the potential to restrict the dispersal of deer. The large number of cattle recently killed in floods and following a “cold snap” in northern Queensland in February 2019 underlines how precarious northern Australia can be for ungulates. The large numbers of animals of all species that perish during prolonged droughts and widespread severe bush fires are also potentially illustrative of the dispersal hazards for deer. We lack information on the rate of recovery of deer species following these catastrophic events in Australia. Interrogating the ecology of deer where they co-exist with different bovids, equids and porcine species under similar climatic and environmental conditions also has the ability to throw light on what to expect across northern Australia. As examples, Kanha National Park in Madhya Pradesh, Baluran National Park in eastern Java and Cobourg/Garig Gunak Barlu National Park in the Northern Territory (where sambar have lived for 150 years with feral banteng, buffalo, pigs and Timor ponies, as well as a range of macropods) all offer opportunities for comparison research.

In northern Australian savannas, termites, fire and herbivory influence forage quality and availability throughout the year; elsewhere, a diverse biomass of native herbivores consumes that forage. In Australia, only relatively few native (marsupial) herbivores and a much higher biomass of introduced large herbivores use the landscape. Recent fire decreases the overall biomass but increases the quality (decreased fibre content and increased crude protein content) of plant material, whereas late dry-season fires result in forage with the highest crude protein content [62]. Interestingly, introduced bovines are strongly attracted to recently burnt areas [63], whereas the response of large native macropods is more variable [62]. 

In Africa, nutrient-poor savannas are dominated by bulk-feeding herbivores, i.e., predominantly large grazers and megabrowsers, whereas regions with nutrient-rich soils generally support greater numbers of mixed-feeding and browsing ruminants [24]. Browsers and mixed feeders obtain additional forage from the canopy layer, e.g., fallen leaves, flowers and fruits. In the short term, these reduce the consumption of seedling and sapling foliage, whereas in the longer term, they can also increase and stabilise mesobrowser abundance [4]. Quantifying these elements of vegetation communities in norther Australia is therefore a key part of the picture.

The coexistence of apparently competitive species co-occurring in the same habitat is a result of resource partitioning [64], and competition can be considered as the bioforce differentiating the use of resources by coexisting species [65]. Predation or different responses of species to environmental factors may also lead to resource partitioning [66]. For example, sympatric species of a similar body size can also compete to avoid predators, where such interaction is characterised as “apparent competition” [67] or competition for “enemy-free space” [68]. These species may consequently try to adopt different escape tactics, sometimes leading to habitat or resource partitioning. In countries with large carnivores like tigers *Panthera tigris*, leopards *P. pardus* and dhole *Cuon alpinus*, competition may be curtailed by predation, keeping the population of competitors below the level at which food resources become limiting [69,70]. It is therefore possible that comparable population regulation occurs in Australia, with dingoes/wild dogs as the predators. Sometimes, predation and competition come together and can affect community assemblage in multiple ways, which can often interact [67,71]. The predation of dispersing animals by dingoes has been anecdotally implicated as a reason why chital have remained relatively closely confined to their original (1886) place of introduction around Maryvale Station north of Charters Towers in Queensland (T. Nevard pers. obs.), although Bentley [72], based on the station owners’ comments, states chital have increased in their numbers since introduction “…despite depredations of dingoes and drought years”. 

Rusa, with mature weights from 75 to 160 kg, and chital, with mature weights from 50 to 110 kg, are much smaller than most other introduced established herbivorous species in tropical northern Australia, and we do not know how they could effectively compete with native species, in particular macropods of a similar size. Sambar, with mature weights from 150 to 350 kg, are more equivalent in size to many of the established introduced feral and domestic herbivorous species, and so there is the potential that they could compete in terms of their feeding behaviour and diet [73]. Another factor that will influence the success or otherwise of the dispersal and long-term establishment of chital, rusa and sambar is whether they are able to ecologically separate themselves. Of relevance to this is that in some locations in India, chital and sambar coexist [40], whereas rusa and chital are allopatric. In Australia, understanding these interwoven relationships and formulating appropriate and cost-effective management regimes will require carefully targeted research.

A precautionary note is that herbivores use and respond to their environment heterogeneously over different spatial scales and modify patterns of landscape elements at numerous levels. For example, their distribution may be influenced at the local scale by soil nutrients and at the landscape scale by aspect and slope. For example, Olff and Ritchie [74] demonstrated that plant diversity, investigated at a small scale, increased in the presence of grazing but at a larger scale decreased. Lundgren and colleagues [31] have also established that the impact on vegetation by native or introduced herbivores is often indistinguishable, underlining the need for strong caution when interpreting apparent vegetation damage in the wider ecological context.

Taking feral horses as an example, a common theme is the desire to maintain horse densities where their ecological damage is minimised and their ecological, economic and cultural benefits are maximised [75]. Research to achieve this balance for most introduced species in Australia has received little attention [75,76]. Successful management requires the determination of a threshold level of population or density below which the impact is benign or acceptable, based on structured measurement of detrimental and beneficial impacts. This threshold level could be zero, but local eradication is rarely achieved, and it may not be the most desirable goal [77].

As a further cautionary note, in the past, goats have been vilified as noxious pests. More recently, because of much better knowledge of their ecological separation from cattle, in a number of circumstances, their past pest status has changed, and they have become highly valued livestock, alongside and complimenting cattle production. Is there the possibility that a commercial environment might evolve in which one or more species of deer could follow the same path?

## 5. Conclusions and Recommendations

Except for the work by Watter and colleagues [43,44,45,46,78] published in 2019 and 2020, there have been very few studies in northern Australia that report ecological interactions between deer species and native and introduced herbivores. As a result of limited literature and empirical studies, it therefore remains unclear as to whether the dispersal and establishment of the three species of deer in northern Australia would effectively compete with similar-sized native species and how deer could impact vegetation communities and the smaller native vertebrate species that inhabit them.

This review has identified some degree of ecological separation of herbivores, but there are clear gaps in the published research looking at how deer coexist or interact with domestic and free-living populations of introduced and native herbivores in tropical Australia’s savannas. We therefore suggest research should be undertaken to address these gaps, focusing on the degree of ecological separation of deer from native and introduced herbivores. Before appropriate informed decisions on the sustainable management of deer in northern Australia can be made, significant additional objective structured scientific investigation is therefore required:What the changes to plant community structure initiated or progressed by deer (grazing, browsing, the removal of seedlings, tree rubbing, etc.) are in different habitats, where populations of native and introduced herbivores are found.Whether deer individually or collectively compete with or negatively impact native herbivores or displace other introduced species.How frequently deer, individually or collectively, are predated by dingoes/wild dogs and crocodiles and whether this predation is a regulator of populations of both deer and dingo/wild dog populations.Whether deer, individually or collectively, will have their populations regulated by current climatic conditions and whether climate change, water sources, fire, floods and drought will be important regulators of deer populations in the future.What is the potential of deer, individually or collectively, to create local industries for meat, skins, velvet or hunting and how these might be engaged by First Nations people, as currently occurs in the Cobourg/Garig Gunak Barlu National Park in the Northern Territory.What the role of deer and other herbivores is as endozoochorous seed dispersers.What the role of deer and other herbivores is as potential reservoirs and vectors for parasites and infectious disease, to determine whether deer are different vectors of disease from other introduced ruminants; andWhether deer have impacts on smaller (<5 kg) native vertebrates and the wide range of invertebrates sharing the same habitats.

## Figures and Tables

**Figure 1 animals-14-01576-f001:**
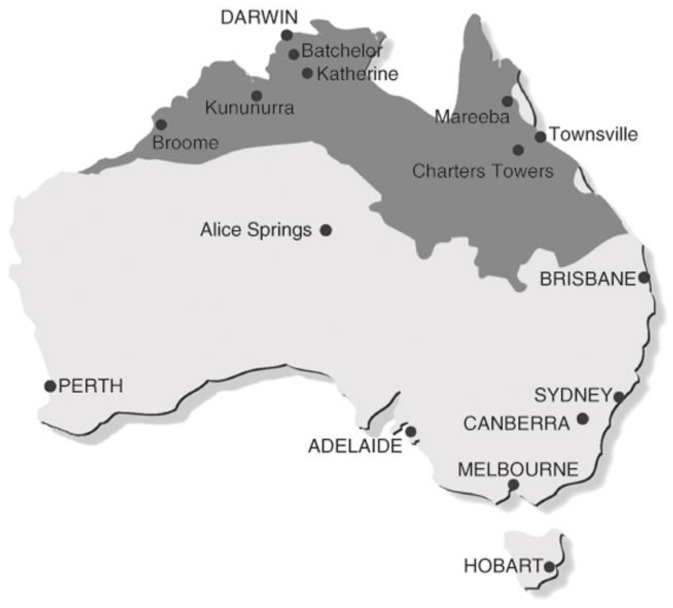
Extent of tropical savannas in northern Australia (after [3]).

**Figure 2 animals-14-01576-f002:**
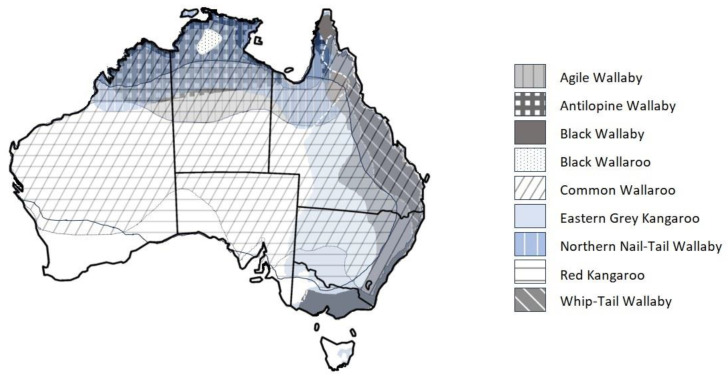
Distribution of macropod species > 5 kg found in northern Australia [32].

**Figure 3 animals-14-01576-f003:**
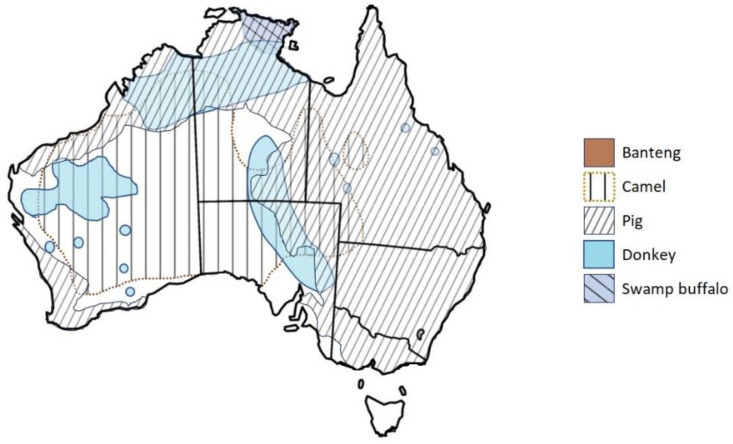
Distribution of the introduced feral pig and herbivorous mammal species > 5 kg found in northern Australia [32]. Not shown are the distributions of feral cattle and horses, as both are widely distributed across northern Australia.

**Figure 4 animals-14-01576-f004:**
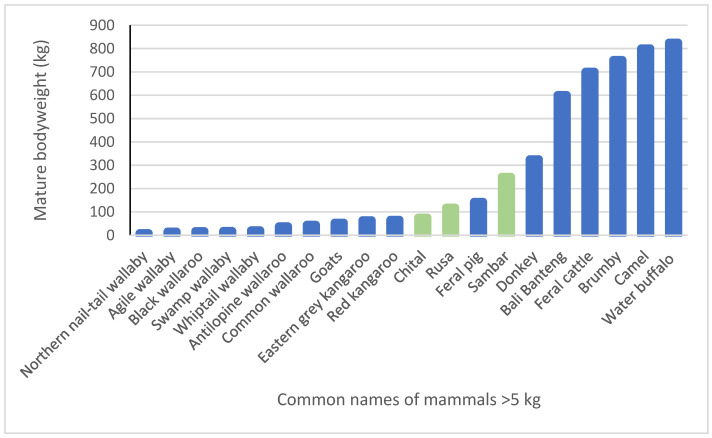
Mean mature bodyweights of native and introduced mammals > 5 kg found in northern tropical Australia; deer species shown in green [32].

**Figure 5 animals-14-01576-f005:**
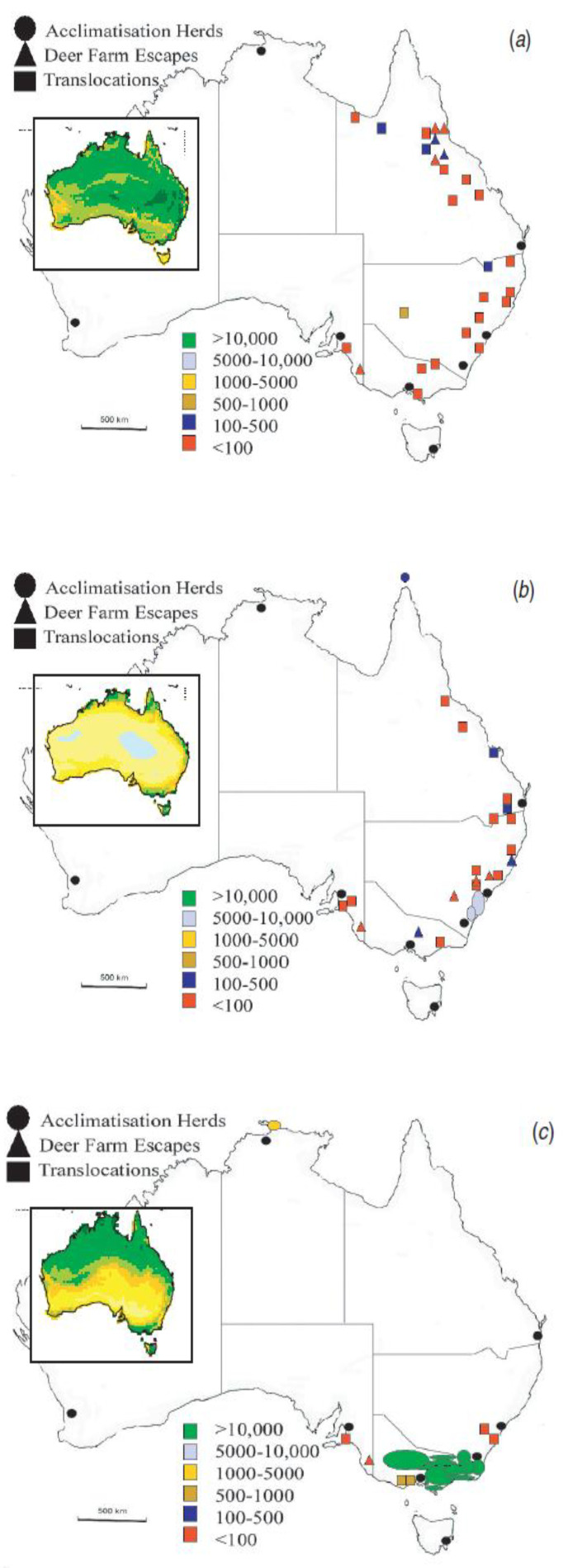
Distribution and abundance of (**a**) chital, (**b**) rusa and (**c**) sambar deer in Australia in 2000 (herd data points not to scale). Insets show the bioclimatic (predicted) distribution of chital, rusa and sambar, respectively (M Bomford, unpublished data). The scale of predicted habitat suitability ranges from pink and dark green, indicating high habitat suitability, to light green and orange, indicating medium habitat suitability, and then to yellow and blue, indicating poor habitat suitability. Taken from Moriarty [34].

**Table 1 animals-14-01576-t001:** Diet, habitat requirements and mature bodyweights of species of native herbivorous vertebrates > 5 kg found in northern tropical Australia (data from [1,17,32]).

Common Name (Mature Weight)	Species Name	Diet and Habitat Requirements
Common wallaroo (28–60 kg)	*Macropus robustus*	Preferential grazer, mainly of grasses; browser of some shrubs; lives in rocky hill country.
Agile wallaby (11–19 kg)	*Macropus agilis*	Most common macropod in tropical coastal Australia; feeds on native grasses, grass roots and some leaves, flowers and fruits.
Red kangaroo (39–92 kg)	*Macropus rufus*	Grazer and browser of grasses, forbs and shrubs; lives in open plains—savannas, open woodlands, arid and semi-arid regions.
Antilopine wallaroo (24–51 kg)	*Macropus antilopinus*	Grazes perennial grasses and some forbs in tropical grasslands with monsoonal eucalypts; at altitudes less than 500 m.
Black wallaroo (13–21 kg)	*Macropus bernardus*	Lives in Arnhem Land rocky escarpments and feeds on grasses, some leaves, flowers, fruits.
Black wallaby (15–20 kg)	*Wallabia bicolor*	Browser that eats shrubs, pasture grasses, agricultural crops, native and exotic vegetation; inhabits thick undergrowth in forests and woodlands, emerging at night to feed.
Whiptail wallaby (15–26 kg)	*Macropus parryi*	Grazer of grasses and monocots near creeks in grasslands and woodlands in central coastal eastern Qld and northern NSW.
Eastern grey kangaroo (42–85 kg)	*Macropus giganteus*	Specialised grazer of a wide variety of grasses across eastern Australia; adaptable but prefers open grassland habitats.
Northern nail-tail wallaby (7–9 kg)	*Onychogalea unguifera*	Feeds on a wide variety of herbs, fruits, succulent plants; will eat grass when herbs are not available; found in arid and sparsely treed plains with tussocks of tough grasses/low shrubs.

Note: native marsupial species > 5 kg that are not included in this table are arboreal, e.g., tree kangaroos *Dendrolagus* spp. and koalas *Phascolarctos cinereus*, as well as species found to the south of northern Australia, e.g., black-striped wallaby *M. dorsalis*, bridled nail-tail wallaby *Onychogalea fraenata* and species that have limited to very limited distributions, such as most rock wallaby *Petrogale* spp. and species living in rainforest, e.g., red-legged pademelon *Thylogale stigmatica*.

**Table 2 animals-14-01576-t002:** Species of introduced herbivorous vertebrates > 5 kg found in northern tropical Australia with a brief description of their diet and habitat requirements. Type of digestive system (DS)—R, ruminant; PR, pseudo-ruminant; HF—hindgut fermenter; O—omnivore; NT—Northern Territory; WA—Western Australia. Deer species highlighted in grey (data from [32]).

Common Name (DS) (Mature Weight)	Species Name	Diet and Habitat Requirements
Feral pig—O (110–175 kg)	*Sus scrofa*	Eats plants, small animals and carcasses; occupies 40% of mainland Australia, associated with most river systems and floodplains, inland drainages and thickly wooded habitats.
Water buffalo—R (450–1200 kg)	*Bubalus bubalis*	Feeds on aquatic grasses, grass-like wetland plants, plus dryland grasses, herbs, pandanus leaves; in the main, a grazing animal on subcoastal plains and river basins between Darwin and Arnhem Land.
Banteng—R (400–800 kg)	*Bos javanicus*	A grazer for c. 200 years in the Cobourg/Garig Gunak Barlu National Park, under First Nations Management; preferred habitat of monsoon forest and associated coastal plain, with freshwater lagoons.
Feral cattle—R (500–900 kg)	*Bos taurus/Bos indicus*	A grazer in a wide range of habitats from forest to semi-desert wetlands.
Goat—R (27–79 kg)	*Capra aegagrus hircus*	A preferential browser that eats leaves, twigs, bark, flowers, fruit, roots and most plant types in pastoral regions, consuming vegetation avoided by sheep or cattle.
Chital—R (50–100 kg)	*Axis axis*	Mainly a grazer but also an intermediate mixed feeder. Populations north of Charters Towers and near Townsville, Barcaldine and Texas in Queensland.
Rusa—R (75–160 kg)	*Rusa timorensis*	Intermediate mixed feeder; will browse depending on season and availability; prefers grassy plains bordered by dense brush or woodlands. Reports from Murulag, Boigu and Saibai islands in the Torres Strait, central Cape York Peninsula, Groote Eylandt, the Gulf Savannah region, around Townsville and Rockhampton and in southern Queensland near Stanthorpe.
Sambar—R (150–350 kg)	*Rusa unicolor*	Intermediate mixed feeder eating a wide variety of grasses, shrubs and tree foliage; prefers forested mountain country and also inhabits open forest with suitable understory cover with gullies. Naturalised under First Nations management in the Cobourg/Garig Gunak Barlu National Park and Western Arnhem Land.
Camel—PR (600–1000 kg)	*Camelus dromedarius*	Preferential browser but will eat most plants and has extraordinary drought tolerance; widely distributed in bushland and sand plains over the arid and semi-arid regions of central Australia.
Donkey—HF (300–350 kg)	*Equus asinus*	Eats grasses, shrubs and tree bark; drought-tolerant, found in the NT and northern and northwest WA.
Feral horse—HF (600–900 kg)	*Equus caballus*	Prefers grassland where drinking water is relatively available; also eats other plants, including tree bark. Occupies over half of Australia, absent from most desert regions and intensively farmed land.

## Data Availability

The scientific literature supporting the conclusions of this article will be made available by the authors on reasonable request.

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
