# Peer review of "The Ecological Separation of Deer and Domestic, Feral and Native Mammals in Tropical Northern Australia—A Review"

_animals, 2024, doi:10.3390/ani14111576_

Round 1
Reviewer 1 Report
Comments and Suggestions for Authors
Dear Authors,
In your review you have examined the ecological and historical factors that led to the formation of the unique guild of native and introduced mammalian herbivores in northern Australia. In particular you have focused on the potential impacts of deer species dispersal in northern Australia and considered the ecological context in which deer, domestic mammals, feral and native mammals coexist, and iyou have identified research gaps that could facilitate appropriate management of expanding deer populations.
In the introduction you described in detail the land cover types and distribution of native and introduced herbivorous mammals and their relative distribution.
The results include a detailed description of the distribution of herbivorous mammals and their feeding ecology. Figures are clear and fine.
Nevertheless, I must point out that your well-written and interesting review could be refined in discussions by also describing the ecology of lagomorphs in Australia. In particular, given the widespread presence of wild rabbits, which have an estimated population of over 200,000,000 individuals in Australia, it would be interesting to comment on:
a) the possible spatial and trophic competition with larger herbivores;
b) whether and how rabbit density may affect predation on coexisting deer species and other mammalian herbivores.
Best regards
Reviewer 2 Report
Comments and Suggestions for Authors
I admit very mixed impressions with this manuscript. On the one hand, it tackles very important topic of relationships between native, non-native and long established, and newly established (mammalian) meso- and mega- herbivores in tropical (northern) Australia. It specifically focuses on thre species of deer, i.e. mesoherbivore browsers to mixed feeders. Despite being based on literature review, it discloses some important topics in Australian ecology. I particularly liked the list of untapped issues at the end of the paper, and appreciate that commercial/productive aspects of having the deer there are addressed.
On the other hand, the text is very imbalanced, some crucial information is missing (or so well hidden that it appears as missing), others are little supported. Modelling results pop up from nowhere, whereas some claims, such as "growing frequency of environmental and climatic disturbances", are repeated several times, but lack a citation support.
The Introduction opens with a long description of tropical Australia (bio)geogeography, which is well written and informative, but should be moved somewhere to Methods section, as Introduction should catch readers attention with something relevant for more general reader. In your case, it clearly is the sad history of Australian megafauna, i.e., its extirpation following human arrival, and factual replacement by feral species, plus comparison with other continents (Australia being similar to South America, and very dissimilar to Africa and South Asia). It all is in the Introduction already, but should be presented in more concise, and comprehensive, way. Last but not least, it is quite unclear why the emhasis is on deer, not even the last sentences of Intro clarify this.
In Methods section, to which the geography details should be moved, I appreciate that you located 800 references, but these are not in references list. I recommend to prepare Appendix, perhaps as xls spreadsheet, in which all these sources would be presented.
A general mistake (e.g., lines 196, 266) are assertions such as "As outlined above..." Good text does not need this, as the ideas follow one from the other.
Finally, in your list of recommendations, I appreciated that you mention effects on vegetation, and effects on small native vertebrates. But what about invertebrates? Here, the impacts, via manipulating plants, may be profound, and deserve to be studied.
Minor points:
125-6: "over two-thirds of the vascular plant 125 genera present in the Northern Territory between 11 and 16°S have global distributions" - I believe that this must be a mistake, or misunderstanding. The plants species may well be "pantropical", or "occurring at several continents", but not globally distributed. They certainly do not grow in northern temperate or boreal regions.
149: Explain "Damuth relationship" for non-expert readers, or skip it entirely.
186: focus animals -> focal animals
Table 1 (and 2): Pay attention to format of the columns, use left-side allignement to avoid the wide spaces, as with some species names.
220: have habitat separation -> display habitat separation
236, and throughout the entire text and tables. Horse, or brumby? You should stick to one term to avoid confusion (although the local Australian word "brumby" might and should be mentioned somewhere).
300: "Climatch analysis (Figure 5)..." - I completely missed an explanation of the analysis. How it works, who did it (you?), how reliable were the results, in terms of models robustness. Must be supplied.
302: "points of liberation" sounds too much value-laden. Use a more neutral term, e.g., "points of release".
395: the phosphorus problem and supplementing for domestic cattle should be backed by a citation
411: Demystifying -> perhaps "Disclosing"?
419: "We remain ignorant of the rate of recovery of deer species..." - I would use more neutral word, rather than "ignorant", but more importantly, there certainly exist population studies of deer populatio recoveries from other parts or the Wordl. Use them to support your argument.
423-5: I appreciate the list of parks "where [deer] co-exist with different bovids, equids and porcine species", but the list is not complete. Why, e.g., Kahna, and not Manas, Kaziranga or Corbett in India, etc. Be either more general, or use very specific examples with citations.
455: Be more cautious regarding dingo as "apex predator". It certanily plays the role in Australia, with its Macropodinae being the largest herbivores, but is hardly comparable with tigers, leopards or lions. More importantly, they were not the apex predators in pre-extinctions Australia, in which much fiercers predators (Thylacoleo, Varanus priscus...) assumed the apex predators role. Also, there is ample evidence, mainly from African savannas, that predators hardly "control" the populations of their prey. They surely modify herbivere guild structure, age and weith distribution, and behaviour, but not necessarily their total numbers.
470-7: "Another factor that will influence the success or otherwise of the dispersal and long-term establishment of chital, rusa and sambar is if they are able to ecologically separate themselves" - I agree with this, but you should note here that chital and sambar coexist across Indian peninsula, and evidently do separate themselves, whereas rusa and chital are allopatric.
Comments on the Quality of English Language
The authors are clearly native speakers, but their text need thorough revision, e.g., with regards to awkward use of some terms.
Round 2
Reviewer 2 Report
Comments and Suggestions for Authors
The manuscript has much improved in the current state. I would personally put even more emphasis on the late pleistocene extinction issue, and on the whole issue that Australian savannas have been defaunated in the past, thus representing "incomplete exosystems" (which is prsently the case of a majority of ecosystems in the World), but these are just personal preferences, and I am trying to avoid my personal preferences in evaluating this, otherwise interesting and useful, paper.